# Investigating the dose-response relationship between music and anxiety reduction: A randomized clinical trial

Danielle K. Mullen[1]*[¤a], Tianle Peng[1], Lauren Stewart[2¤b], Adiel Mallik[3¤a], Frank A. Russo[3,4]

**1** Department of Psychology, Goldsmiths, University of London, London, England, United Kingdom, **2** School of Psychology, University of Roehampton, London, United Kingdom, **3** LUCID Therapeutics, Toronto, Ontario, Canada, **4** Department of Psychology, Toronto Metropolitan University, Toronto, Ontario, Canada

¤a Current address: Department of Psychology, Toronto Metropolitan University, Toronto, Ontario, Canada
¤b Current address: School of Psychology, University of Roehampton, London, United Kingdom
* daniellekmullen@gmail.com

## Abstract

Anxiety is one of the most frequently reported mental health conditions worldwide, yet access to effective treatments such as medication and cognitive behavioral therapy (CBT) remains limited due to cost, time, and potential side effects. Music-based digital therapeutics, particularly when combined with auditory beat stimulation (ABS), may offer a complementary approach to mainline anxiety treatment by offering acute relief of anxiety symptoms. Prior research suggests that music combined with ABS provides greater anxiety relief than music alone or a pink noise control. This study examined whether this advantage over pink noise could be replicated, as well as whether music with ABS demonstrated a dose-response relationship—operationalized as time spent listening—in the acute relief of anxiety among individuals with moderate trait anxiety who are taking medication to manage their symptoms. We also assessed changes in affect as a secondary outcome. A total of 1,310 participants were recruited via Prolific and completed a pre-screening survey. Of these, 144 eligible participants were randomly assigned to one of four groups: 24-minute pink noise (control group), 12-minute music with ABS, 24-minute music with ABS, or 36-minute music with ABS. Anxiety and affect were measured before and after the intervention using the STICSA and PANAS, respectively. All music with ABS conditions resulted in greater reductions in anxiety and negative affect compared to the control, replicating earlier findings. The largest reduction in negative affect was observed in the 36-minute condition, which was significantly greater than reduction in the 12-minute condition, suggesting a dose-response effect. These findings support music with ABS as a possible addition to existing anxiety treatments, especially when access to common behavioral health interventions is limited. Future studies should aim to

**Data availability statement:** The de-identified data that supports the findings of this study has been made available on the Open Science Framework at this link: https://osf.io/wpefy/.

**Funding:** This work was partially funded by LUCID, the developer of the music curation system used in the study. Additional support came from Goldsmiths, University of London, by whom author LS was employed during the time of study conceptualization and data collection. Author FAR holds a leadership position (Chief Science Officer) at LUCID; this conflict of interest was disclosed to all collaborators and to the relevant ethics boards prior to data collection. FAR's contributions to this article were undertaken solely in his academic role. Neither funder participated in or influenced the study design, data collection, data analysis, interpretation of the results, or preparation of the manuscript.

**Competing interests:** Adiel Mallik was employed as a Research Scientist by LUCID between 2022 and 2024. Frank Russo has served as an advisor for LUCID since 2018 and as Chief Science Officer since 2021. Frank Russo has been granted stock options, which may qualify him to financially benefit from commercial applications of the technology considered here. This does not alter our adherence to PLOS ONE policies on sharing data and materials.

increase the generalizability of the findings and further investigate the dose-effect of music on anxiety reduction. This study was retrospectively registered on ISRCTN (ISRCTN47181782).

---

## 1. Introduction

Anxiety is one of the most frequently reported mental health conditions around the world, with over 301 million individuals living with anxiety in 2019 [1]. More recently, there was a significant increase in reported anxiety disorders around the world during and after the start of the COVID-19 pandemic [2]. For those with generalized anxiety disorder, the symptoms experienced can be both cognitive, such as having a sense of dread, or somatic, such as experiencing trembling. For those with Social Anxiety Disorder, symptoms can also include being fearful of social situations and subsequently avoiding those situations. For those with Panic Disorder, the cognitive symptoms can include the feeling of impending doom while the somatic symptoms can include a racing heart. In all cases, these symptoms can take a serious toll on one's quality of life, including by negatively impacting one's ability to function socially and causing heightened negative, and lowered positive affect [3].

Today there are several leading treatments for anxiety management, with two of the most frequently sought being medication and psychotherapy. Among the most common medications prescribed for anxiety symptoms are selective serotonin reuptake inhibitors [4], and benzodiazepines [5]. While these pharmaceutical interventions have been found to be effective at reducing anxiety [6], this course of symptom management has several drawbacks. One of the commonly discussed disadvantages to using medication for anxiety management is the potential for negative physical side effects, especially with long-term benzodiazepine use [7]. When benzodiazepine use is stopped abruptly, patients can experience intense withdrawal symptoms, such as insomnia, with the severity depending on how long they have been taking the medications [8]. Benzodiazepines pose even more risk to those with a history of addiction disorders, as they have the potential of being addictive when used for more than 4 consecutive weeks [9]. While other commonly prescribed medications such as selective serotonin reuptake inhibitors (SSRIs) do not share these potential side effects and are therefore considered by many to be a safer medication alternative [10,11], SSRIs still have their own frequently reported physical side effects, including nausea and sexual dysfunction [12]. Further, SSRIs have been reported to cause emotional blunting, often described as limiting one's range of emotions [13] or causing a general feeling of apathy [14]. Finally, SSRIs take between 2 and 4 weeks to start having the intended anxiolytic effect on the patient taking them, leaving a gap of time when patients are still needing therapeutic aids. These adverse side effects can make clinicians less inclined to prescribe these medications [7] and patients less inclined to adhere to their scheduling [15].

In contrast, increasing evidence exists to support the use of cognitive behavioral therapy (CBT) for the treatment of anxiety [16]. CBT often aims to alter maladaptive

thoughts and problematic behaviors of patients with anxiety, which in turn can improve these patients' overall quality of life [17]. In a 2012 review of meta-analyses examining the efficacy of CBT, it was consistently found that CBT was largely effective in reducing anxiety symptoms, including improving "secondary" symptoms such as sleep dysfunction [18]. Evidence has also shown CBT to still be effective when delivered online [19], which would suggest an increased accessibility for those who are unable (or unwilling) to see a practitioner in person at a clinic. Newer approaches to CBT such as "third wave mindfulness-based CBT", may be more accessible and have also been found to be effective in treating anxiety in a variety of populations [20,21]. Additionally, some patients opt to pursue a combination of medication and psychotherapy to manage their anxiety, though a 2014 meta-analysis found insufficient evidence to support the combination of medication and psychotherapy over medication alone [22].

Despite the existing empirical evidence that supports the use of CBT and its variants, there are limitations. Because all forms of CBT involve discussing and facing the things that make a person anxious, the first few sessions can be particularly uncomfortable, especially for those who tend to be avoidant of the things that make them anxious, which may result in discontinuation [23]. Additionally, consistent attendance at CBT sessions is needed for the best outcomes [24], which poses an obstacle for those that are not able to commit to this level of adherence. This impact is further exacerbated by the cost of this type of treatment where there is no state funded healthcare, with a single CBT session often costing more than $100 USD [25]. All these obstacles to accessible, safe, and economically attainable treatment for anxiety have led to a demand for complementary or stepped-care therapeutic options for symptom management.

One such complementary approach involves the use of music-based digital therapeutics. Listening to music has already been found to be popular amongst a range of populations for managing negative mood states and improving affect (including relieving anxiety), such as those in adolescence and early adulthood [26], undergraduate students [27,28], military veterans [29], and older adults [30]. This is perhaps because negative emotions can be effectively down-regulated through music [31]. Further, music interventions have been found to be effective in altering one's emotional state in various clinical populations, including those with mild cognitive impairment [32], Alzheimer's disease [33], depression [34] and generalized anxiety disorder [35]. For patients with anxiety, both music listening and music making have been found to abate somatic symptoms, including elevated blood pressure and heart rate, as well as cognitive symptoms, such as worry and nervousness [36]. Music listening has even been found to be more effective at reducing preoperative anxiety than midazolam, a frequently used benzodiazepine for preoperative anxiety [37]. Other advantages to using music listening for anxiety reduction include the fact that music can be self-administered and may be more widely accessible and cost effective than medication and CBT [29].

Music listening is not, however, a panacea for the alleviation of anxiety and there are several considerations regarding its use. Among these are the use of the iso-principle and the additional benefits of combining auditory beat stimulation (ABS) with the music being used. First, the iso-principle suggests that music's efficacy in regulating emotion is enhanced when the initial music selections convey mood and arousal levels that approximate the patient's current state [38]. Over the course of a session, the music selections are made strategically in a manner that moves the patient's mood and arousal closer to the target. For example, if a patient was in a high arousal state (such as feeling anxious) and wanting to feel calm, using music for this goal would theoretically be more effective if the music was upbeat to start then gradually became slower and more relaxed. Previously it has been reported that the use of the iso principle can result in more significant reductions in pain and tension [39]. While the evidence supporting the use of the iso principal is limited [40], it is widely adopted by practicing music therapists and has been shown in a case study to be greatly effective in improving mood management [38].

Second, auditory beat stimulation (ABS) is typically created by presenting two sine waves of slightly different (neighboring) frequency, resulting in a difference tone, which may be perceived as a distinct tone or amplitude modulation [41]. When two sine waves are presented with one in each ear, binaural beats are perceived. Alternatively, two of the amplitude modulated beat signals can be presented to either one or both ears, resulting in a monaural beat. These beats can be

in the delta (1–4 Hz), theta (4–8 Hz), alpha (8–13 Hz), beta (14–30 Hz), or gamma (30–50 Hz) ranges. Previous studies have shown listening to ABS alone to be effective in reducing anxiety, especially when the ABS has a frequency falling in and around the theta range [42,43]. As for combining music with ABS, studies have found that listening to music with ABS can result in greater reduction in heart rate than listening to music alone [44], greater reduction in somatic anxiety than pink noise and greater reduction in cognitive anxiety than music alone, ABS alone, or pink noise [45]. Mallik & Russo (2022) also found that a 24-minute use of a digital therapeutic involving music combined with ABS significantly reduced anxiety in patients with moderate trait anxiety as compared to listening to pink noise alone for the same length of time [45].

While these findings are encouraging, there are still several questions about the combination of music listening with ABS on anxiety reduction that remain, such as whether a dose-response relationship can be seen. Many studies on music and anxiety reduction have had participants listen to music for somewhere between 15 and 60 minutes per session [46], while a 2019 meta-analysis of studies using binaural beats for anxiety reduction reported on interventions lasting anywhere between 4 and 130 minutes [47]. Although a previous study found anxiety to be significantly reduced after a single 24-minute session of listening to music with ABS [45], there were no comparisons made to music with ABS of other lengths of time.

The current study set out to replicate the finding of Mallik and Russo (2022), that music with ABS would be superior to pink noise in its abilities to reduce anxiety, and to extend this, in exploring whether the reduction in anxiety and improvement in affect experienced when listening to music with ABS differs according to the exposure duration using a between groups design. An exploratory aim of the study was to determine if there were any changes in arousal and valence experienced by the participants after listening to either music with ABS or pink noise. A previous study using a music with binaural beats intervention for anxiety reduction found significant reductions in participants' self-reported level of anxiety, as determined by changes in State Trait Anxiety Inventory scores, after the participants listened to their assigned audio for 60 minutes [44]. Further, a previous meta-analysis reported the optimal treatment time for music-based treatments to be between 20 and 30 minutes [47]. Based on these findings, it was hypothesized that the longer participants listened to music, the greater their reduction in anxiety and improvement in affect would be, operationalized as changes in self-report scores on anxiety and affect scales before and after listening to their assigned audio intervention.

Additionally, it was hypothesized that all groups listening to music would have greater reduction in anxiety and improvement in affect compared to the group listening to pink noise. This was based on the prior finding that participants who listened to music with ABS had reductions in both cognitive and somatic anxiety and improvements in positive affect compared to participants who listened to a pink noise control [45].

## 2. Methods

### 2.1. Ethics statement

This study received approval from the Ethics Committee of the Department of Psychology at Goldsmiths, University of London and the Toronto Metropolitan University Research Ethics Board. All participants were over 18 years old and gave written informed consent prior to participating in this study.

### 2.2. Design

This study used a between-subjects design and was conducted entirely online. The independent variable was the length of time spent listening to a music playlist with theta ABS: short exposure (approximately 12 minutes, ranging between 11-14.5 minutes), medium exposure (approximately 24 minutes, ranging between 23.5-29.5 minutes), or long exposure (approximately 36 minutes, ranging between 35.5-42.5 minutes). In the prior study on which the current study is based (Mallik & Russo, 2022), audio interventions of approximately 25 minutes were used, and this was found to be an effective length of time spent listening to music with ABS for anxiety reduction. The 12- and 36-minute exposure times were determined by adjusting the 24-minute audio by 12-minute intervals. The 12-minute sessions incorporated 4 tracks, and

the 36-minute sessions incorporated 12 tracks. As each track was approximately 3-minutes in length, the overall duration was approximately 12-, 24-, or 36-minutes in length. As a control, one additional group listened to pink noise for exactly 24 minutes.

Pink noise has a power spectral density proportional to 1/f (frequency)—roughly a 3dB decrease in power per octave—and is known to be present in a range of naturally occurring settings [48], such as in the flow of sand falling in an hourglass and the flow of water in the Nile River [49]. Previous meta-analyses on the use of music for anxiety reduction have emphasized the importance of using an active control group to compare the effect of music on anxiety reduction with [36,46]. To this end, using pink noise rather than silence helps to control for the possibility of a placebo effect that may be seen when having some groups listen to music while others sit in silence [45].

Although the study was originally planned to have a fully balanced design with 12-, 24-, and 36-minute pink noise groups, piloting with 36 participants revealed poor adherence to the 36-minute pink noise group (50%). Adherence was determined by comparing the amount of time spent on the study to the length of the assigned audio. For example, if someone was assigned to a 36-minute audio but only spent 20 minutes on the study, they were considered to have not adhered to the audio. Additionally, a participant provided feedback after completing the survey and reported feeling nauseous and dizzy while listening to the audio. As a result, it was decided to remove the 36-minute pink noise group from the study to avoid any other unintended adverse effects. We then also removed the 12-minute pink noise group in order to have a more balanced study design, leaving the 24-minute pink noise audio as the sole control group. Removing the 12-minute pink noise made it so that each condition was being compared to a single control group.

As one of the goals of this study was to replicate the findings of Mallik & Russo (2022), it is important to note that while the intervention in the earlier study was app-based, the current study used an integrated web-based solution for both survey data and deployment of the digital-therapeutic intervention. This change was made to give participants a more seamless experience, as some participants in Mallik & Russo (2022) experienced technical issues with app installation leading to frustration that may have influenced baseline affect scores.

## 2.3. Self-report measures

The primary outcome was reduction in state anxiety. As in the prior study [45], this was determined by participant's scores on the State Trait Inventory for Cognitive and Somatic Anxiety (STICSA) before and after listening to their assigned playlist. Studies on the reliability and validity of the STICSA have found that it is able to reliably detect changes in somatic anxiety symptoms [50], has validity as a measure of both cognitive and somatic trait anxiety and more strongly measures anxiety symptoms than the State-Trait Anxiety Inventory [51]. The secondary outcome was change in affect, as determined by participants ratings on the Positive and Negative Affect Scale (PANAS) before and after listening to their assigned playlist [45]. The PANAS consists of 20 self-report items and was chosen for its' high reliability and validity, as well as its' sensitivity to mood fluctuation when used in short-term contexts [52]. Both the STICSA and PANAS were used in the prior study which the current is based on [45].

## 2.4. Treatment conditions

Participants were randomly assigned to one of the 4 groups, 3 experimental (music with ABS) and 1 control (pink noise), using the Qualtrics randomizer algorithm. Once participants were invited to complete the study, they could access the Qualtrics page at their convenience. This study was registered retrospectively on ISRCTN (ISRCTN47181782), where the study trial is available.

## 2.5. Participants

Participants were recruited through Prolific, an online platform for digitally connecting researchers to participants anonymously (www.prolific.co). The inclusion criteria were that participants had to be currently taking anxiolytic medication

(to ensure a clinical diagnosis) and to have moderate trait anxiety (STICSA trait cognitive score between 17.1 and 26.6, STICSA trait somatic score between 16.9 and 22.4, as previously defined as the STICSA moderate trait anxiety threshold scores [5]). The need to meet the threshold for moderate trait anxiety was informed by the previous study on which the current is based [45] which found that listening to music with ABS was more effective for reducing anxiety in those with moderate trait anxiety than those with high trait anxiety. Recruitment took place from July 17th, 2023, to August 12th, 2023.

Recruitment began with pre-screening 1310 participants on Prolific for eligibility, with 144 meeting the inclusion criteria and consenting to the study (for demographic information, see Table 1, for medication information, see Table 2, and for alternate coping methods used, see Table 3). The most common reason for exclusion was not meeting the threshold for moderate trait anxiety. Of the 144 participants in the final sample, there were 41 in the 12-minute music with ABS group, 33 in the 24-minute music with ABS group, 34 in the 36-minute music with ABS group, and 36 in the 24-minute pink noise control group (Fig 1).

**Table 1. Participant demographic information.**

| Treatment | Gender Distribution | Mean Age (years) (± SD) |
|---|---|---|
| Control | 13 males (36%), 23 females (63.89%) | 35.91 (11.60) |
| 12 min music | 18 males (43.90%), 23 females (56.10%) | 38.44 (11.55) |
| 24 min music | 12 males (36.36%), 21 females (58.33%) | 37.06 (11.08) |
| 36 min music | 13 males (38.24%), 20 females (58.83%), 1 N/A (3%) | 38.76 (13.64) |

**Table 2. Participant medication information.**

| Treatment | SSRI | SNRI | BB | BZ | AH | AP | NDRI | AC | NaSSA | 5-HT-RA | ST | TA | Other | SSRI and BZ | Multiple medications |
|---|---|---|---|---|---|---|---|---|---|---|---|---|---|---|---|
| Control | 18 | 1 | 3 | 5 | 0 | 0 | 0 | 0 | 0 | 1 | 0 | 2 | 4 | 0 | 2 |
| 12 min music | 18 | 1 | 1 | 9 | 0 | 0 | 0 | 0 | 0 | 2 | 1 | 2 | 3 | 2 | 2 |
| 24 min music | 16 | 0 | 5 | 2 | 0 | 1 | 0 | 0 | 0 | 0 | 0 | 0 | 4 | 2 | 3 |
| 36 min music | 14 | 0 | 3 | 8 | 0 | 0 | 0 | 0 | 0 | 0 | 0 | 0 | 4 | 1 | 3 |
| Total # of participants | 66 | 2 | 12 | 24 | 0 | 1 | 0 | 0 | 0 | 3 | 1 | 4 | 15 | 5 | 10 |

Abbreviations: SSRI = Selective Serotonin Reuptake Inhibitor, SNRI = Serotonin Norepinephrine Reuptake Inhibitor, BB = Beta Blocker, BZ = Benzodiazepine, AH = Antihistamine, AP = Antipsychotic, NDRI = Norepinephrine-Dopamine Reuptake Inhibitor, AC = Anticonvulsant, NaSSA = Noradrenergic and specific serotonergic antidepressant, 5-HT-RA = Serotonin 5-HT1A Receptor Agonist, ST = Stimulant, TA = Tricyclic Antidepressants.

Source Questionnaire: Anxiety Coping Methods Questionnaire.

Timepoint: Beginning of the Study (Pre-treatment).

**Table 3. Participant alternate coping methods information.**

| Treatment | Physical Exercise | Breathing Exercises | Mindfulness | Physical + Breathing | Physical + Mindfulness | Other Combinations |
|---|---|---|---|---|---|---|
| Control | 1 | 3 | 3 | 7 | 2 | 5 |
| 12 min music | 2 | 6 | 2 | 3 | 5 | 2 |
| 24 min music | 4 | 7 | 3 | 1 | 4 | 3 |
| 36 min music | 9 | 5 | 4 | 1 | 3 | 4 |
| Total # of participants | 16 | 21 | 12 | 12 | 14 | 14 |

Source Questionnaire: Anxiety Coping Methods Questionnaire.

Timepoint: Beginning of the Study (Pre-treatment).

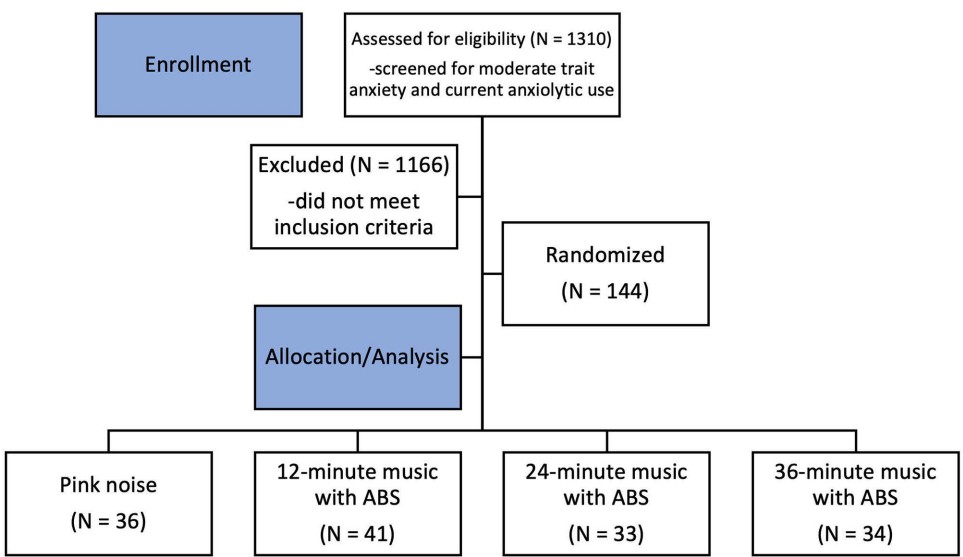

**Fig 1. Consort participant flow diagram.**

## 2.6. Sample size

The sample size was determined by an *a priori* power analysis based on two previous studies that investigated the relationship between binaural beats and anxiety [43,45]. We conducted an *a priori* t-test (difference between independent means) power analyses using G-Power 3.1.9. The prior studies had a combined Cohen's D of 0.80, indicating that a minimum of 26 participants per group would be required to achieve a power of 0.80 at a significance of $p = 0.05$. We recruited additional participants above the minimum of 26 to account for the 25–50% participant dropout rate that is present in many online research studies [53,54].

## 2.7. Materials

This experiment was conducted online using Qualtrics, an online survey software (https://www.qualtrics.com). A brief anxiety coping methods questionnaire was used to confirm that participants were currently taking an anxiolytic, which anxiolytic they were taking, as well as to assess any other anxiety coping methods used by participants, such as other medications, including cannabis, or alternative techniques such as breathing techniques, exercise, and meditation.

The auditory interventions used in this study were instrumental music with nature sounds, combined with theta-band ABS. These audios were composed and provided by LUCID Inc, and were drawn from the same database of stimuli tested by Mallik & Russo (2022). To create the digital therapeutic intervention, LUCID uses an Affective Music Recommendation System (AMRS), which draws on the iso principle by meeting participants at their current emotional state and moving them to their desired state.

To make use of the AMRS curation, the Self-Assessment Manikin (SAM) was completed by participants. Doing so allowed us to identify their baseline and post-intervention emotional states with respect to valence and arousal. The SAM is a set of pictorial representations of 5 levels of valence, one's degree of positive or negative emotion ranging from pleasant to unpleasant, and 5 levels of arousal, one's energy level ranging from calm to excited [55]. The SAM has been used in previous studies to measure overall affect [56], general valence and arousal [57], emotional status [58], and pleasantness and arousal in response to a range of sounds, such as people laughing and sports events [59]. Further, the SAM has been found to have high reliability and validity as a measure of emotion [60]. For participants assigned to an experimental

group, their responses to the SAM corresponded to the specific playlist they were given in their time group. The SAM has possible scores from 1 to 5 on both valence and arousal, resulting in 25 possible combinations of valence and arousal that participants could have potentially indicated. Thus, 25 different playlists for each time point in the music with ABS groups were created, resulting in the use of a total of 75 music with ABS playlists.

### 2.8. Procedures

Participants were pre-screened so that only individuals currently taking an anxiolytic with moderate trait anxiety were able to access the study. Prolific screened participants on their anti-anxiety medication use. 1310 participants that self-identified on Prolific as taking an anti-anxiety medication were invited to complete the STICSA. If they scored in the moderate trait anxiety range, they were invited to complete the study on Qualtrics. After being directed to a designated Qualtrics page and consenting to participate in the study, participants were randomly assigned to one of the four groups using the Qualtrics randomizer algorithm.

To start the study, participants completed the Anxiety Coping Methods Questionnaire, PANAS, and STICSA. Participants were then asked to indicate their current pleasure and arousal levels using the SAM, and subsequently were presented with the playlist corresponding to their arousal and valence level for their randomly assigned amount of time. Although participants in the control group also completed the SAM, their response did not have any bearing on the playlist they were presented with as there was only one pink noise audio. All participants were asked to close their eyes and wear headphones while listening to their assigned audio.

After participants finished listening to their assigned playlist, they all once again completed the SAM, PANAS, and STICSA. As a check of compliance and engagement, participants were asked if they had completed the audio session with headphones along with if they had been distracted during the audio session, and if so, to what extent (i.e., "not distracted at all" to "so distracted that I (they) could not focus on the audio").

### 2.9. Statistical methods

There were 144 participants in the current study. Pre-post listening changes were determined for the cognitive and somatic dimensions of the STICSA, the positive and negative dimensions of the PANAS, and the valence and arousal dimensions of the SAM. This procedure yielded six change scores for each participant: change in cognitive anxiety, change in somatic anxiety, change in positive affect, change in negative affect, change in valence, and change in arousal.

Pairwise Fisher randomization resampling tests (2000 iterations), also known as permutation tests, were run to generate p-values associated with differences in change scores across groups. Permutation tests are able to control the type I error rate for multiple comparisons, as this non-parametric approach makes no assumptions about the underlying distribution of the data that are common in other inferential statistical tests [61–64].

## 3. Results

87 participants self-identified as female, 56 as male, and 1 preferred not to identify. The participants ranged in age from 19 to 73, with the average being 37.6 years (SD = 11.9 years). Most participants were British (N = 56) followed by American (N = 37), though participants were recruited from around the world. Each participant was compensated financially for their time, and completion of the study was determined by comparing the amount of time participants spent on the study to the length of their assigned audio.

### 3.1. Replication: Changes in anxiety

The changes in anxiety in the pink noise control group versus 24-minute music with ABS group can be seen, both for the current and prior (Mallik & Russo, 2022) study, in Fig 2 (changes in cognitive anxiety) and Fig 3 (changes in somatic anxiety).

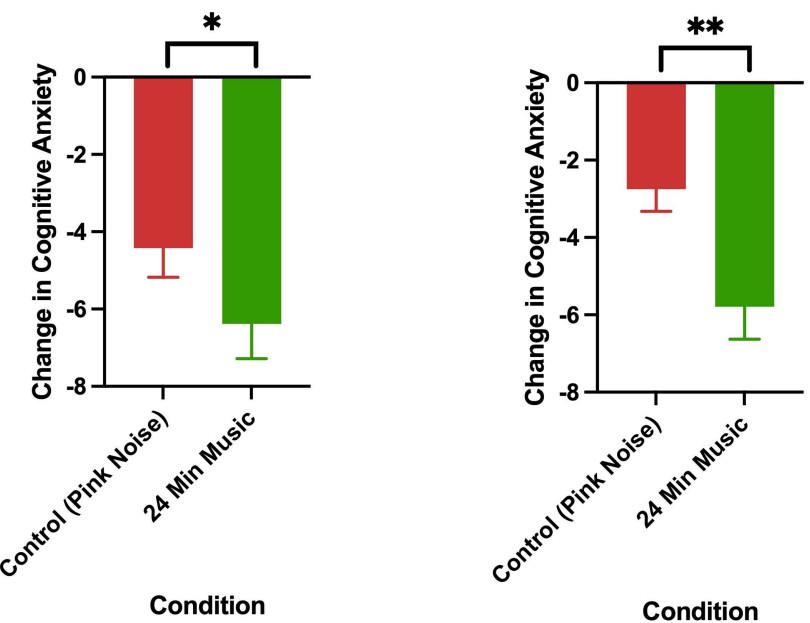

**Fig 2. Change in cognitive anxiety as measured by the STICSA (post-pre) in A) prior study (Mallik & Russo, 2022) p<0.05 for control (24-minute pink noise) vs 24-minute music with ABS comparison, and B) current study p<0.01 for control (36-minute pink noise) vs 24-minute music with ABS comparison.** Error bars depict the standard error of mean. * denotes p<0.05, ** denotes p<0.01, **** denotes p<0.0001.

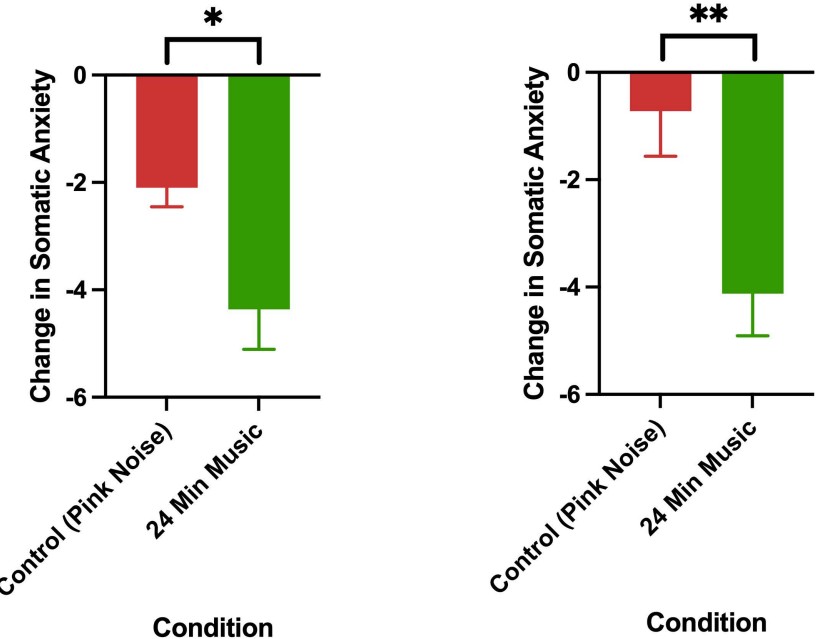

**Fig 3. Change in somatic anxiety as measured by the STICSA (post-pre) in A) prior study (Mallik & Russo, 2022), p<0.05 for control (24-minute pink noise) vs 24-minute music with ABS comparison, and B) current study, p<0.01 for control (36-minute pink noise) vs 24-minute music with ABS comparison.** Error bars depict the standard error of mean. * denotes p<0.05, ** denotes p<0.01, **** denotes p<0.0001.

Permutation tests showed the 24-minute music with ABS group had higher reductions in both cognitive and somatic anxiety than the pink noise control group in both the current and prior study, and these differences were all found to be significant with a medium effect size (Tables 4 and 5). For both cognitive and somatic anxiety, stronger statistical evidence was found in the current study than in the prior one, as evidenced by the current study's lower p-values (current cognitive: p = 0.004, prior cognitive: p = 0.05; current somatic: p = 0.003, prior somatic: p = 0.04).

The mean cognitive anxiety reductions (± standard deviation) for each group were as follows: Current Control (Pink Noise) = -0.72 (5.05), Current 24-Minute Music with ABS = -5.79 (4.85), Prior Control (Pink Noise) = -4.42 (3.83), Prior 24-Minute Music with ABS = -6.38 (4.14). The mean baseline cognitive anxiety scores (± standard deviation) for each group were as follows: Current Control (Pink Noise) = 19.33 (6.16), Current 24-Minute Music with ABS = 20.52 (5.15), Prior Control (Pink Noise) = 23.19 (7.91), Prior 24-Minute Music with ABS = 21.64 (7.29). The mean somatic anxiety reductions (± standard deviation) for each group were as follows: Current Control (Pink Noise) = -2.75 (3.45), Current 24-Minute Music with ABS = -4.12 (4.51), Prior Control (Pink Noise) = -2.10 (1.64), Prior 24-Minute Music with ABS = -4.36 (3.49). The mean baseline somatic anxiety scores (± standard deviation) for each group were as follows: Current Control (Pink Noise) = 16.08 (4.72), Current 24-Minute Music with ABS = 17.33 (4.85), Prior Control (Pink Noise) = 19.17 (6.80), Prior 24-Minute Music with ABS = 17.21 (6.03).

### 3.2. Replication: Changes in affect

The comparison of changes in affect in the pink noise control group versus 24-minute music with ABS group in the prior study (Mallik & Russo, 2022) and the current study are visualized in Fig 4 (changes in positive affect) and Fig 5 (changes in negative affect).

Permutation tests showed that while the prior study had an increase in positive affect in the 24-minute music with ABS group, and this difference was significant from the change in positive affect in the pink noise group (p = 0.005), the difference between the change in positive affect between the 24-minute music with ABS group and pink noise group in the current study was not significant. Additionally, in the current study, both the pink noise group and the 24-minute music with ABS group experienced a reduction in positive affect rather than an increase.

Conversely, in the current study, the 24-minute music with ABS group had significantly higher reductions in negative affect than the pink noise control group (p = 0.0085) with a medium effect size (Table 6). This finding was also inconsistent with the prior study, where there was no significant difference in negative affect reduction between the 24-minute music with ABS group and the pink noise group.

The mean positive affect changes (± standard deviation) for each group were as follows: Current Control (Pink Noise) = -1.72 (6.18), Current 24-Minute Music with ABS = -3.15 (6.36), Prior Control (Pink Noise) = -3.89 (6.99), Prior 24-Minute

**Table 4. Additional statistical information for significant comparisons for cognitive state anxiety in moderate trait anxiety participants.**

| Comparison | Exact p-value | Effect size (Cohen's d) | 95% CI for Effect Size |
|---|---|---|---|
| Control vs. 12 min music | 0.017 | 0.45 | 0-0.91 |
| Control vs. 24 min music | 0.004 | 0.72 | 0-0.96 |
| Control vs. 36 min music | 0.05 | 0.50 | 0-0.95 |

**Table 5. Additional statistical information for significant comparisons for somatic state anxiety in moderate trait anxiety participants.**

| Comparison | Exact p-value | Effect size (Cohen's d) | 95% CI for Effect Size |
|---|---|---|---|
| Control vs. 12 min music | 0.029 | 0.50 | 0-0.91 |
| Control vs. 24 min music | 0.003 | 0.71 | 0-0.96 |
| Control vs. 36 min music | 0.0045 | 0.68 | 0-0.95 |

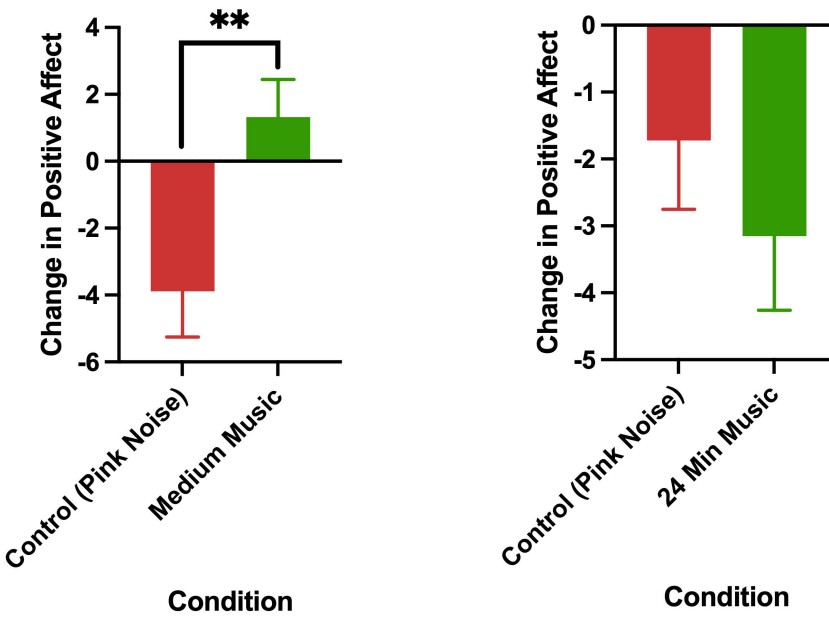

**Fig 4. Change in positive affect as measured by the PANAS (post-pre) in A) prior study (Mallik & Russo, 2022), p < 0.01 for control (24-minute pink noise) vs 24-minute music with ABS comparison, and B) current study Error bars depict the standard error of mean.** * denotes p < 0.05, ** denotes p < 0.01, **** denotes p < 0.0001.

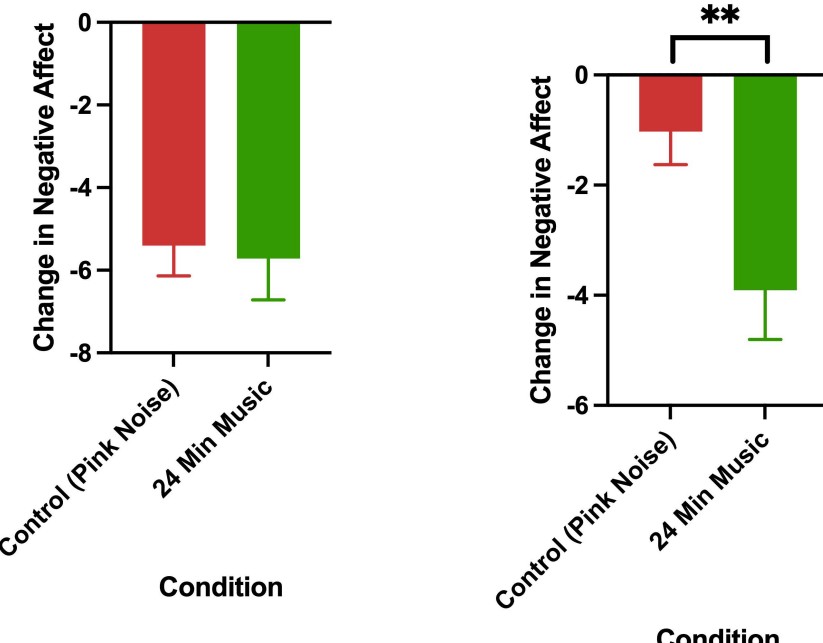

**Fig 5. Change in negative affect as measured by the PANAS (post-pre) in A) prior study (Mallik & Russo, 2022), B) Current Study, p < 0.01 for control (36-minute pink noise) vs 24-minute music with ABS comparison Error bars depict the standard error of mean.** * denotes p < 0.05, ** denotes p < 0.01, **** denotes p < 0.0001.

**Table 6. Additional statistical information for significant comparisons for negative affect in moderate trait anxiety participants.**

| Comparison | Exact p-value | Effect size (Cohen's d) | 95% CI for Effect Size |
|---|---|---|---|
| Control vs. 12 min music | 0.017 | 0.55 | 0-0.91 |
| Control vs. 24 min music | 0.0085 | 0.65 | 0-0.96 |
| Control vs. 36 min music | 0.000017 | 0.97 | 0-0.95 |
| 12 min music vs 36 min music | 0.024 | 0.55 | 0-0.92 |

Music with ABS = +1.32 (5.28). The mean baseline positive affect (± standard deviation) for each group were as follows: Current Control (Pink Noise) = 25.78 (6.30), Current 24-Minute Music with ABS = 27.49 (6.34), Prior Control (Pink Noise) = 28.23 (8.27), Prior 24-Minute Music with ABS = 27.46 (7.05). The mean negative affect reductions (± standard deviation) for each group were as follows: Current Control (Pink Noise) = -1.03 (3.61), Current 24-Minute Music with ABS = -3.91 (5.12), Prior Control (Pink Noise) = -5.40 (3.83), Prior 24-Minute Music with ABS = -5.71 (4.58). The mean baseline negative affect (± standard deviation) for each group were as follows: Current Control (Pink Noise) = 17.06 (7.47), Current 24-Minute Music with ABS = 17.09 (6.52), Prior Control (Pink Noise) = 23.07 (8.81), Prior 24-Minute Music with ABS = 20.06 (9.41).

### 3.3. Extension: Changes in anxiety

The changes in cognitive and somatic anxiety by treatment group in the current study are visualized in Figs 6 and 7, respectively.

Permutation tests showed all music with ABS groups had higher reductions in both cognitive and somatic anxiety than the pink noise control group, and these differences were all found to be significant. The greatest significant difference was between the control and 24-minute music with ABS group in both cognitive and somatic anxiety reduction (cognitive: p = 0.004; somatic: p = 0.003).

The mean cognitive anxiety reductions (± standard deviation) for each group were as follows: Control (Pink Noise) = -2.75 (3.45), 12-Minute Music with ABS = -4.44 (3.96), 24-Minute Music with ABS = -5.79 (4.85), 36-Minute Music with ABS = -4.76 (4.50). The mean baseline cognitive anxiety scores (± standard deviation) for each group were as follows: Control (Pink Noise) = 19.33 (6.16), 12-Minute Music with ABS = 21.03 (6.53), 24-Minute Music with ABS = 20.52 (5.15), 36-Minute Music with ABS = 20.85 (6.04). The mean somatic anxiety reductions (± standard deviation) for each group were as follows: Control (Pink Noise) = -0.72 (5.05), 12-Minute Music with ABS = -2.90 (3.59), 24-Minute Music with ABS = -4.12 (4.51), 36-Minute Music with ABS = -4.03 (4.64). The mean baseline somatic anxiety scores (± standard deviation) for each group were as follows: Control (Pink Noise) = 16.08 (4.72), 12-Minute Music with ABS = 17.17 (4.96), 24-Minute Music with ABS = 17.33 (4.85), 36-Minute Music with ABS = 17.65 (5.00).

### 3.4. Extension: Changes in affect

The changes in positive and negative affect by treatment group in the current study are visualized in Figs 8 and 9, respectively.

Permutation tests showed that none of the changes in positive affect between groups were significant, though they had a medium effect size (Table 7). Conversely, all music with ABS groups had significantly higher reductions in negative affect than the pink noise control group, with the greatest significance being between the control and 36-minute music with ABS group (p < 0.001). Additionally, there was a significant difference in the reduction of negative affect between the 12-minute music with ABS group and the 36-minute music with ABS group (p = 0.024).

The mean positive affect reductions (± standard deviation) for each group were as follows: Control (Pink Noise) = -1.72 (6.18), 12-Minute Music with ABS = -1.71 (4.84), 24-Minute Music with ABS = -3.15 (6.36), 36-Minute Music with

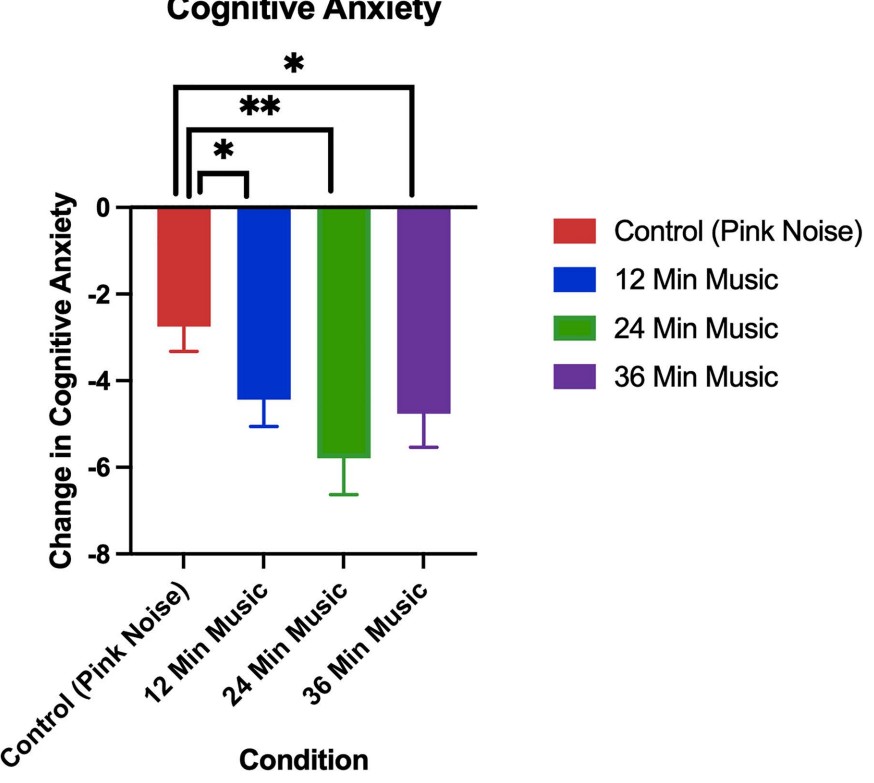

**Fig 6. Change in cognitive anxiety as measured by the STICSA (post-pre).** Error bars depict the standard error of mean. $p < 0.05$ for control vs 12-minute music with ABS comparison, $p < 0.01$ for control vs 24-minute music with ABS comparison, $p < 0.05$ for control vs 36-minute music with ABS comparison. * denotes $p < 0.05$, ** denotes $p < 0.01$, **** denotes $p < 0.0001$.

ABS = -0.26 (5.85). The mean baseline positive affect scores (± standard deviation) for each group were as follows: Control (Pink Noise) = 25.78 (6.30), 12-Minute Music with ABS = 26.03 (7.16), 24-Minute Music with ABS = 27.49 (6.34), 36-Minute Music with ABS = 26.76 (7.75). The mean negative affect reductions (± standard deviation) for each group were as follows: Control (Pink Noise) = -1.03 (3.61), 12-Minute Music with ABS = -2.98 (3.45), 24-Minute Music with ABS = -3.91 (5.12), 36-Minute Music with ABS = -5.47 (5.39). The mean baseline negative affect scores (± standard deviation) for each group were as follows: Control (Pink Noise) = 17.06 (7.47), 12-Minute Music with ABS = 16.83 (6.83), 24-Minute Music with ABS = 17.09 (6.52), 36-Minute Music with ABS = 17.74 (6.92).

### 3.5. Extension: Changes in arousal

The changes in arousal by treatment group are visualized in Fig 10.

Permutation tests showed that the reduction in arousal was significantly greater in the 24-minute music with ABS group than the pink noise, 12-minute music with ABS, or 36-minute music with ABS groups. The mean arousal reductions (± standard deviation) for each group were as follows: Control (Pink Noise) = 0 ± 1.24, 12-Minute Music with ABS = -0.32 ± 0.85, 24-Minute Music with ABS = -0.73 ± 0.91, 36-Minute Music with ABS = -0.18 ± 1.03.

### 3.6. Extension: Changes in valence

The changes in valence by treatment group are visualized in Fig 11.

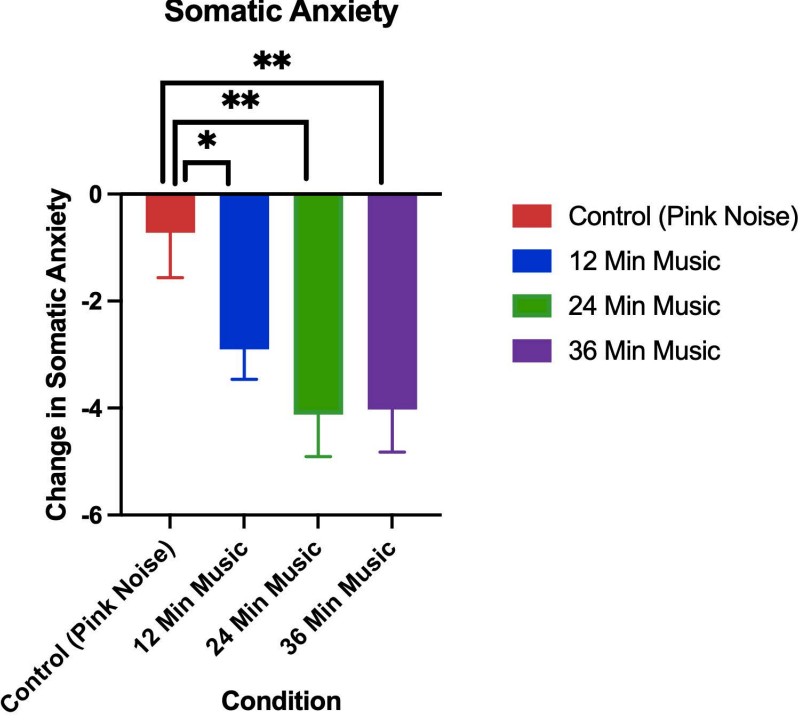

**Fig 7. Change in somatic anxiety as measured by the STICSA (post-pre).** Error bars depict the standard error of mean. $p < 0.05$ for control vs 12-minute music with ABS comparison, $p < 0.01$ for control vs 24-minute music with ABS comparison, $p < 0.01$ for control vs 36-minute music with ABS comparison. * denotes $p < 0.05$, ** denotes $p < 0.01$, **** denotes $p < 0.0001$.

Permutation tests showed that the increase in valence was significantly greater in all music with ABS groups compared to the pink noise group. The mean valence increases (± standard deviation) for each group were as follows: Control (Pink Noise) = 0 ± 0.96, 12-Minute Music with ABS = 0.39 ± 0.86, 24-Minute Music with ABS = 0.64 ± 0.96, 36-Minute Music with ABS = 0.59 ± 0.93.

## 4. Discussion

There were two primary aims of the current study: (1) to replicate the findings of Mallik and Russo (2022) concerning the effects of a music-based digital therapeutic on anxiety relative to pink noise, and (2) to investigate a potential dose-response relationship on reduction in anxiety. A secondary aim was to investigate a potential dose-response relationship on affect. Additionally, an exploratory aim was to determine if there were any changes in arousal and valence experienced by participants.

### 4.1. Replication of Mallik and Russo (2022)

Consistent with our hypotheses, participants in the 24-minute music with ABS group experienced greater reductions in both cognitive and somatic anxiety than those in the pink noise group, and these differences were all shown to be significant with a medium effect size, which replicates the findings of Mallik and Russo (2022). This replication helps to support the notion that listening to music with ABS is an evidence-based method for anxiety management and can be more effective than listening to other auditory stimuli used to regulate anxiety, namely pink noise. The findings are more nuanced when it comes to the PANAS, our secondary outcome measure. Whereas the earlier study found that music increased positive affect and pink noise reduced it, the current study showed a reduction of positive affect in

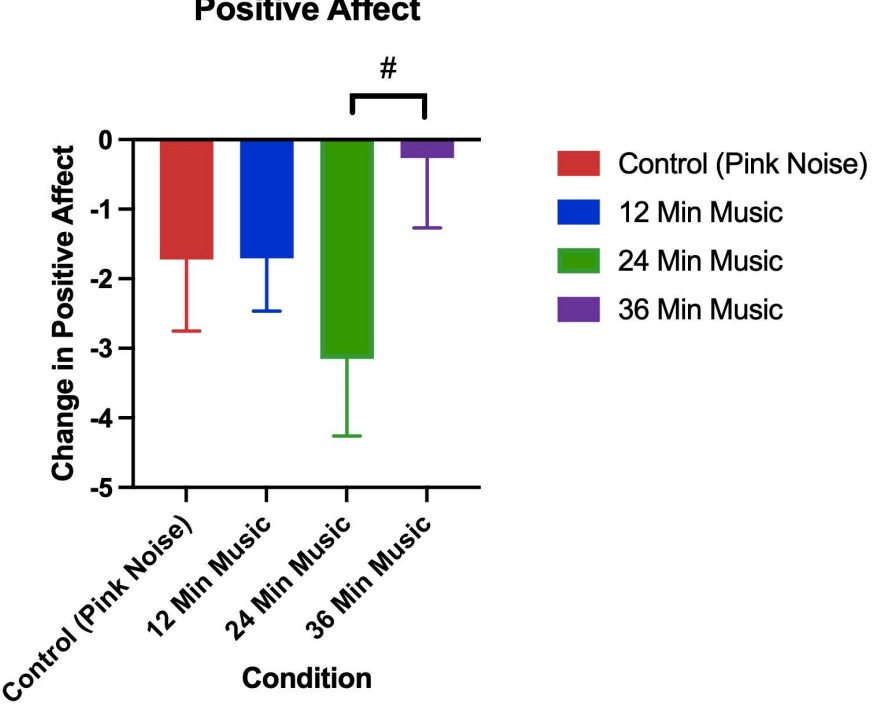

**Fig 8. Change in positive affect as measured by positive and negative affect scale (post-pre).** Error bars depict the standard error of mean. p < 0.1 for 24-minute music with ABS vs 36-minute music with ABS comparison. # denotes p < 0.1,* denotes p < 0.05, ** denotes p < 0.01, **** denotes p < 0.0001.

both groups. In addition, while Mallik and Russo (2022) found that pink noise and music with ABS both led to a similar reduction in negative affect, the current study showed a greater reduction in negative affect for music with ABS over pink noise. In short, relative to Mallik and Russo (2022), the between-groups difference in positive affect was diminished, whereas the between-groups difference in negative affect was amplified, i.e., music produced a greater reduction in negative affect than pink noise.

Differences in the mode of intervention delivery across studies may help to explain this discrepant pattern of findings. First, the current study's seamless web interface reduces procedural friction that was experienced in the prior study (no downloading or troubleshooting), which plausibly attenuated previous sources of irritation caused by technical difficulties. This interpretation is supported by inspection of the baseline data—specifically, baseline negative affect was higher in the earlier study than the current study for pink noise (M = 23.07 vs. M = 17.06) as well as for music with ABS (20.06 vs. 17.09). However, caution is warranted when interpreting differences in PANAS results across studies. While the contextual explanations we offer are plausible, they remain speculative and would require further testing for systematic evaluation.

### 4.2. Extension of Mallik and Russo (2022)

Turning to the extension of the prior study, the 24-minute and 36-minute music with ABS groups yielded the greatest numerical reduction in both cognitive and somatic anxiety compared with the 24-minute pink noise group. Although no significant differences were found between any of the music with ABS groups for cognitive anxiety, the 24-minute music group had a higher reduction and higher statistical significance and greater effect size (Table 4) in its pink noise comparison than the other music with ABS groups. Similarly, in the case of somatic anxiety, although the 24- and 36-minute music with ABS groups were not statistically significant compared to the 12-minute music with ABS group, the 24-minute and

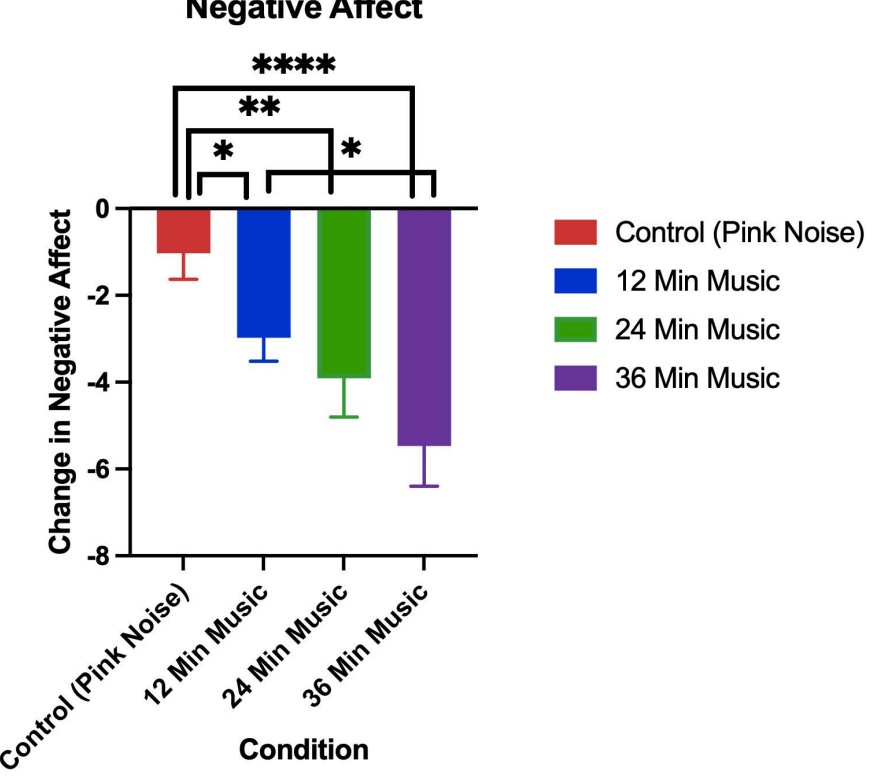

**Fig 9. Change in negative affect as measured by positive and negative affect scale (post-pre).** Error bars depict the standard error of mean. $p < 0.05$ for control vs 12-minute music with ABS comparison, $p < 0.01$ for control vs 24-minute music with ABS comparison, $p < 0.0001$ for control vs 36-minute music with ABS comparison, $p < 0.05$ for 12-minute music with ABS vs 36-minute music with ABS comparison.* denotes $p < 0.05$, ** denotes $p < 0.01$, **** denotes $p < 0.0001$.

**Table 7. Additional statistical information for significant comparisons for positive affect in moderate trait anxiety participants.**

| Comparison | Exact p-value | Effect size (Cohen's d) | 95% CI for Effect Size |
|---|---|---|---|
| 24 min music vs. 36 min music | 0.06 | 0.47 | 0-0.97 |

36-minute music with ABS groups had a higher reduction and higher statistical significance and greater effect size (Table 5) in its pink noise comparison than the 12-minute music with ABS group. This suggests that there is potentially a dose effect for cognitive and somatic anxiety if statistical power were to be increased with greater sample size.

All groups (including pink noise) experienced an unexpected reduction in positive affect, and none of the differences between groups were found to be significant, though the difference between the 24-minute and 36-minute music with ABS groups was approaching significance and had a medium effect size. This suggests that it may take longer than 24-minutes for participants to start to enjoy the music and hence increase positive affect. Conversely, while all groups experienced a reduction in negative affect, all music with ABS groups had higher reductions in negative affect than pink noise, with all of these differences being significant. Additionally, the 36-minute music with ABS group was shown to have a significantly greater reduction in negative affect compared to that of the 12-minute music with ABS group, along with a large effect size, suggesting 36-minute music with ABS was the most effective dose for negative affect reduction when compared to 24-minutes of pink noise. These negative affect findings replicate and extend those of Mallik and Russo (2022), revealing

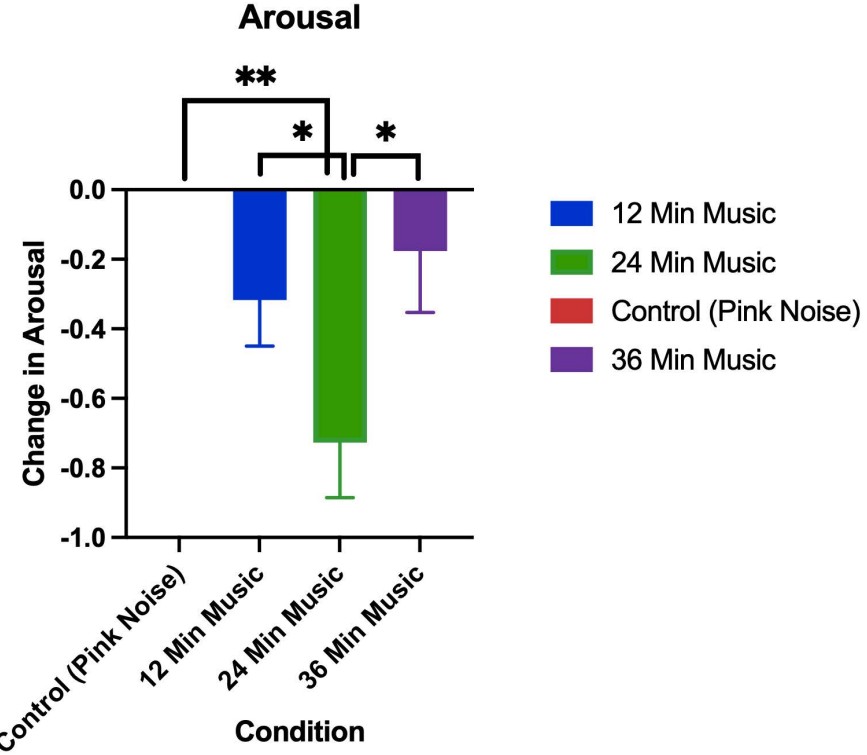

**Fig 10. Change in arousal as measured by SAM (post session minus pre-session).** Error bars are standard errors of means. p < 0.01 for control vs 24-minute music with ABS comparison, p < 0.05 for 12-minute vs 24-minute music with ABS comparison, p < 0.05 for 24-minute vs 36-minute music with ABS comparison. * denotes p < 0.05, ** denotes p < 0.01.

that longer exposures are most effective with respect to negative affect reduction, i.e., that there is a dose effect when using music with ABS for negative affect reduction.

Interestingly, the arousal and valence findings relate very well with the somatic and cognitive anxiety findings. For arousal, we see that all three music groups significantly decreased arousal compared to the pink noise group. In addition, we find that the 24-minute music with ABS group had significantly higher arousal decreases compared to the 12-minute and 36-minute music with ABS groups. We find a similar pattern in cognitive anxiety with the 24-minute music group having the highest mean anxiety reduction compared to the other music groups although not statistically significant. Indeed, the cognitive anxiety portion of the STICSA greatly reflects negative affect as well as arousal [65]. This suggests that the reduction in negative affect and arousal that we have observed is linked with the reductions we see in cognitive anxiety.

In the case of valence, we see a pattern of all of the music with ABS groups having significantly higher valence compared to the pink noise group. The 24-minute music with ABS group has a higher mean valence compared to the other music with ABS groups although this was not statistically significant. There is some evidence to suggest that diminished processing in positive valence systems is associated with generalized anxiety disorder and social anxiety disorder [66]. But it is important to note there is a stronger association between negative valence system dysfunction and anxiety disorders [66]. It would appear that arousal, valence, and negative affect are associated with the decreases we observed in anxiety. Further research is needed to definitively determine whether a causal link between arousal, valence, negative affect, and anxiety exists, and what behavioral/neurological mechanisms drive these behaviors.

All of the participants in the current study were pre-screened to confirm they were currently taking an anxiolytic. Many of the participants reported taking some form of an SSRI (N = 66), which can lead to emotional blunting [13]. This blunting

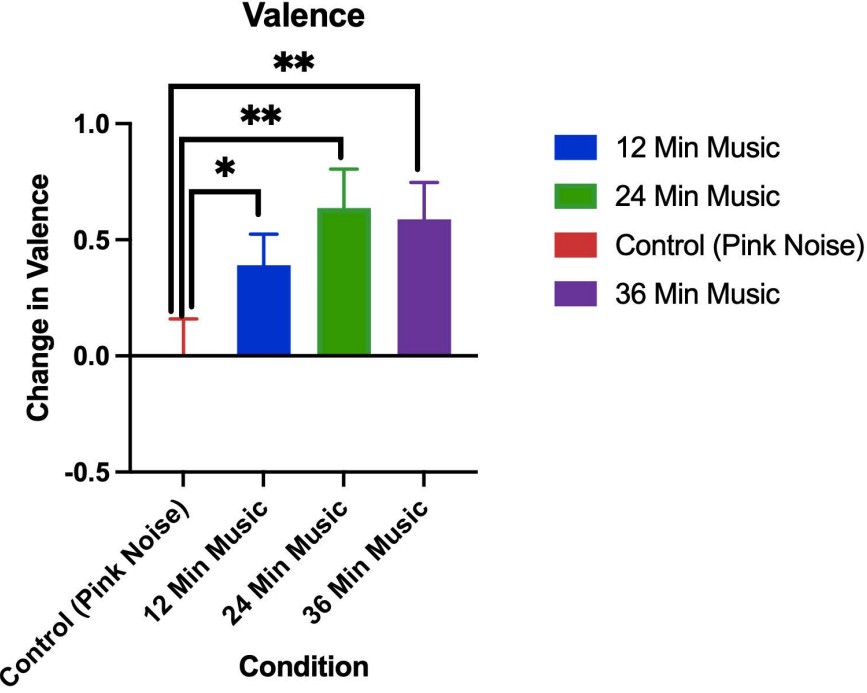

**Fig 11. Change in valence as measured by SAM (post session minus pre-session).** Error bars are standard errors of means, p < 0.05 for control vs 12-minute music with ABS comparison, p < 0.01 for control vs 24-minute music with ABS comparison, p < 0.01 for control vs 36-minute music with ABS comparison. * denotes p < 0.05, ** denotes p < 0.01.

effect on the participant's emotions is a possible explanation for why the longest duration of music with ABS (36-minutes) led to the greatest reduction in negative affect. In particular, individuals with emotional blunting may require more time to be influenced by emotion that is conveyed by the music. Carrying this logic forward, it is possible that individuals with less emotional blunting may reach optimal benefit with shorter exposures, but this is a question that demands further research.

Despite the observed decreases in somatic and cognitive anxiety, reductions in negative affect and arousal, and increases in valence, we observed no significant increases in positive affect. This could be due to several reasons.

The participant population had moderate trait anxiety and were also on anxiolytic medication. Anxiolytic medication such as SSRIs can cause emotional blunting which is indicative of diminished neural processing of both rewarding and aversive stimuli [67], reduced activation in frontal cortex and reward circuits [67,68], and decreased dopaminergic function in prefrontal regions critical for positive emotions [68]. Outside of anxiolytic effects, given that our participant population had moderate trait anxiety, our participants' baseline dopaminergic function may have been compromised through chronic stress and anxiety [68]. Other work shows that participants on anxiolytics also have enhanced sensitivity to anxiety-reducing interventions due to their stress response systems being already active [69]. This may have created a situation where music and theta ABS reduced anxiety via parasympathetic activation and theta entrainment without activating the dopaminergic pathways needed for positive affect enhancement [70,71].

This study provides further support for the contention that music listening can be used to reduce state anxiety in individuals with a clinically significant level of trait anxiety. The findings are of particular relevance in the treatment of individuals coping with anxiety, as this study demonstrated the complementary benefits of music listening to help reduce anxiety in

addition to taking an anxiolytic. In other words, these findings encourage the utilization of music listening among those prescribing treatments for anxiety so that music listening is prescribed to help with managing anxiety in addition to taking an anxiolytic to improve treatment outcomes. Music listening may also be prescribed in cases where patients are unable or unwilling to engage in other modalities such as CBT or medication, where other modalities have been ineffective, or where patients have had adverse reactions to other modalities, as engaging in music listening in these scenarios is likely better than the patient receiving no treatment at all.

### 4.3. Limitations

There were several limitations to the current study. First, regarding compliance with the protocol and immersion in the music, participants were asked to close their eyes and wear headphones while listening to their assigned audio to ensure they were paying attention to the audio. Because this experiment was conducted entirely online, there was no way to ensure complete compliance to these instructions, meaning participants may not have been fully immersed in the music. This is a limitation because some of the benefits of listening to music with ABS may have been mitigated if the participants were not fully immersed. In particular, any effects due to the ABS would not have been present in the absence of head-phones. Along these lines, while conducting this experiment in participants' natural environments enhances the external validity of the findings of this study by simulating real-world conditions in which patients with anxiety would be likely to use this intervention, this may have in turn reduced the internal validity of the study as there was less experimental control as compared to conducting the study in a laboratory setting.

In an attempt to only include individuals with clinically significant anxiety, participants were required to be taking anti-anxiety medication, which was verified at multiple points during their study participation. However, we did not verify a formal diagnosis of anxiety disorder for each participant. Additionally, no data was collected on the dosage of medications that the participants were taking. Because of this, we cannot say what combination of anti-anxiety medication and music listening is ideal for anxiety reduction. Additionally, because of the lack of dosage information, we cannot speak to the extent of emotional blunting that may have been occurring in the participants.

Another limitation of the current study is that the effect of binaural beats may be influenced by participant factors. Research shows that ABS stimulation works by entraining brain activity to specific frequencies, but this entrainment is more effective when individuals naturally exhibit oscillations near the stimulating frequency. If endogenous activity at the target frequency (e.g., 4 Hz theta) is absent or weak, the external auditory beat may fail to induce the desired neural syn-chrony or cognitive and behavioral effects. Other participant-specific factors—such as music preferences—can influence ABS efficacy [72], which we did not account for in the current study.

A final limitation is that this study used only one pink noise audio, 24-minutes in length. This is a limitation because we did not have a control for each experimental time point to be compared.

## 5. Conclusion

The findings of the current study suggest that listening to music with ABS is more effective than listening to pink noise in treating state anxiety in people living with moderate trait anxiety. While there was a trend suggestive of a dose effect on cognitive and somatic anxiety, the trend was non-significant and the numerical peak was at 24-minutes rather than 36-minutes. However, we did find clear evidence of a dose effect for negative affect reduction, with the 36-minute expo-sure leading to a significantly greater reduction than the 12-minute exposure. These findings are important in potentially helping practitioners incorporate music as an alternative to traditional therapeutic channels or part of a stepped-care therapeutic solution for supporting individuals living with anxiety. Further research is needed to better understand the dose effect of music-based digital therapeutics, such as the ideal dose for positive affect improvement, but these findings are encouraging for those with moderate anxiety who are looking an affordable, accessible, and non-medication-based method for relief.

## Supporting information

**S1 Checklist. CONSORT checklist.**
(DOC)

## Author contributions

**Conceptualization:** Danielle K. Mullen, Tianle Peng, Lauren Stewart, Adiel Mallik, Frank A. Russo.

**Data curation:** Danielle K. Mullen, Tianle Peng.

**Formal analysis:** Danielle K. Mullen, Adiel Mallik.

**Investigation:** Danielle K. Mullen, Tianle Peng.

**Methodology:** Danielle K. Mullen, Tianle Peng, Lauren Stewart, Adiel Mallik, Frank A. Russo.

**Project administration:** Lauren Stewart.

**Software:** Danielle K. Mullen, Tianle Peng, Adiel Mallik.

**Visualization:** Danielle K. Mullen.

**Writing – original draft:** Danielle K. Mullen.

**Writing – review & editing:** Danielle K. Mullen, Lauren Stewart, Adiel Mallik, Frank A. Russo.

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
