## [Decision Letter · Decision Letter 0]

10 Dec 2024

PMEN-D-24-00371

Investigating the dose-response relationship between music and anxiety reduction: A randomized clinical trial

PLOS Mental Health

Dear Dr. Mullen,

Thank you for submitting your manuscript to PLOS Mental Health. After careful consideration, we feel that it has merit but does not fully meet PLOS Mental Health’s publication criteria as it currently stands. Therefore, we invite you to submit a revised version of the manuscript that addresses the points raised during the review process.

We look forward to receiving your revised manuscript.

Kind regards,

Bochra Nourhene Saguem, M.D.

Academic Editor

PLOS Mental Health

Journal Requirements:

1. When you resubmit, please ensure you have classified the submission as a "clinical trial" rather than a "research article.

Additional Editor Comments (if provided):

Reviewers' comments:

Reviewer's Responses to Questions

**Comments to the Author**

1. Does this manuscript meet PLOS Mental Health’s publication criteria?

Reviewer #1: Partly

Reviewer #2: Partly

2. Has the statistical analysis been performed appropriately and rigorously?

Reviewer #1: Yes

Reviewer #2: I don't know

3. Have the authors made all data underlying the findings in their manuscript fully available (please refer to the Data Availability Statement at the start of the manuscript PDF file)?

Reviewer #1: Yes

Reviewer #2: Yes

4. Is the manuscript presented in an intelligible fashion and written in standard English?

Reviewer #1: Yes

Reviewer #2: Yes

Reviewer #1: The authors conduct an experiment on the effects of the duration of music listening on reductions in anxiety, with respect to pink noise. While the manuscript is well written, I have concerns about the conclusions made by the authors, given the limitations of the experimental controls used.

Major comments:

1. The design of the experiment may be insufficient for the conclusions drawn for the paper: where in one condition – ABS + music, in the other condition – pink noise only, with the variable for conditions being the amount of time a participant was exposed to the music stimuli. My main concern is that it does not appear to be a valid control, in that other aspects of music not related to ABS or iso-principles are not controlled for. For example, could the effect be simply an effect of listening to music (as the authors pointed out in the introduction) over pink noise, where the duration is reflective of the calming effects of music? Note that research on mood regulation in music by Saarikallio has pointed out 3 ways that negative emotions (including anxiety) may be downregulated – discharge (or catharsis, which may be what the iso-principle is associated with), solace, and diversion. Any of these could be explaining the significant results observed in the study. A better control would be to have musical elements without either ABS, or iso-principles as part of the composition, as secondary controls. However, this may be a bit too late for this study, given that the design cannot be changed, so this should be made clear in the limitations, and some of the conclusions of the study on the efficacy of ABS and the iso-principle may need to be reduced, in favour of the general effect of music listening.

Saarikallio, S. H. (2008). Music in mood regulation: Initial scale development. Musicae scientiae, 12(2), 291-309.

2. Pink noise appears to exacerbate physiological reactions (nausea)? This could be an additional confound of the study, where participants in the pink noise condition may be experiencing additional negative elements (e.g., boredom, annoyance, nausea) that may be explaining the muted results (see above). While this paradigm may have been a replication of previous work, it still does not offer strong evidence for the effect of ABS on anxiety reduction due to these issues with the experiment (namely with the lack of good controls).

Minor comments:

Why would listening to pink noise lead to poor adherence in the planned control conditions for 12 and 36 minutes? Can the authors provide more explanation here perhaps?

May I ask for more information about the auditory stimuli (listening examples)

The listening period is quite long for an online study. How did the authors ensure compliance, such as the participants not doing something else while the music was playing.

P values cannot = 0, but should be expressed as < .001.

In the discussion (p17), the 3 study aims are mentioned, but only 2 are elaborated on immediately after.

The paper lacks a limitations section

Reviewer #2: It was a pleasure reviewing this manuscript for publication. My full review is presented in the attached pdf - please see attachment for more details, comments and recommendations. Thank you for your submission.

what does this mean?). If published, this will include your full peer review and any attached files.

**Do you want your identity to be public for this peer review?** For information about this choice, including consent withdrawal, please see our Privacy Policy

Reviewer #1: No

Reviewer #2: No

---

## [Decision Letter · Decision Letter 1]

11 Jul 2025

PMEN-D-24-00371R1

Investigating the dose-response relationship between music and anxiety reduction: A randomized clinical trial

PLOS Mental Health

Dear Dr. Mullen,

Thank you for submitting your manuscript to PLOS Mental Health. After careful consideration, we feel that it has merit but does not fully meet PLOS Mental Health’s publication criteria as it currently stands. Therefore, we invite you to submit a revised version of the manuscript that addresses the points raised during the review process.

The manuscript has been evaluated by three reviewers, and their comments are available below.

The reviewers have raised a number of major concerns. They request improvements to the reporting of methodological aspects of the study, for example, regarding the control group and justifications for diverging from methods used in the trial that is being replicated. The reviewers also note concerns about the statistical analyses presented. Please note some of their comments are available as attachments.

Could you please carefully revise the manuscript to address all comments raised?

We look forward to receiving your revised manuscript.

Kind regards,

Jenna Scaramanga

Staff Editor

PLOS Mental Health

Journal Requirements:

Additional Editor Comments (if provided):

Reviewers' comments:

Reviewer's Responses to Questions

**Comments to the Author**

Reviewer #2: (No Response)

Reviewer #3: (No Response)

Reviewer #4: (No Response)

publication criteria?

Reviewer #2: Partly

Reviewer #3: Yes

Reviewer #4: (No Response)

3. Has the statistical analysis been performed appropriately and rigorously?

Reviewer #2: Yes

Reviewer #3: Yes

Reviewer #4: No

4. Have the authors made all data underlying the findings in their manuscript fully available (please refer to the Data Availability Statement at the start of the manuscript PDF file)?

Reviewer #2: Yes

Reviewer #3: Yes

Reviewer #4: Yes

5. Is the manuscript presented in an intelligible fashion and written in standard English?

Reviewer #2: Yes

Reviewer #3: Yes

Reviewer #4: No

Reviewer #2: Please see attachment for full review and comments.

Reviewer #3: Overall, the manuscript describes very interesting research related to music and the potential as alternative support people with mental health issues. The methods and materials are quite complicated and can be confusing from the way they were described throughout the manuscript. However, previous reviewers have extensively reviewed the theoretical and technical parts of the research which I mostly agree and do not wish to repeat.

I just have a few questions/ concerns as the following:

1- How were the conditions (short, medium, long) verified from each participants? Since it was administered online, does the application actually measure the actual time of the participants listening to the music? If not and it was only assumed from the group assignment, this may be another limitation.

2- The results have a lot going on and thus, confusing. Consider explaining in one section under methodology: the statistical tests, pre and post analysis and all the variables (IVs and DVs from questionnaire, SAM, PANAS, STICSA and so on) you're measuring or comparing.

3- Can the effect of ABS also be influenced by participant's factors or situations not measured in the study? Participants reaction to the music may well be influenced by their social status, current life events, works, physical health, the place and time they listen to the music etc. which may temper with the results. This should also be discussed thoroughly.

4- In terms of representation, can the results be generalized to any particular sampling population? I am not familiar with Prolific for the sampling or what areas or countries it covers

Reviewer #4: Line 268: "Recruitment was stopped once we reached this minimum sample requirement." However, the flowchart shows that each arm enrolled at least 33 participants, which is over 25% more than the stated minimum of n = 26. Furthermore, the sample sizes across treatment groups are not balanced, ranging from 33 to 41 participants—a difference of more than 20%.

What statistical method was used to calculate the sample size?

Lines 244–250 would be more appropriately placed in the Results section .

P-values should be adjusted for multiple comparisons.

Tables require reformatting:

Treatment arms should be presented as columns.

For Table 1: add % for gender; show SD for age.

For outcome measures:

Show mean (SD) for both baseline and change from baseline.

Avoid displaying unnecessary technical details such as "two-sided", degrees of freedom, etc., in the column headings.

Post-hoc power calculations are not meaningful and should be omitted.

what does this mean?). If published, this will include your full peer review and any attached files.

**Do you want your identity to be public for this peer review?** For information about this choice, including consent withdrawal, please see our Privacy Policy

Reviewer #2: No

Reviewer #3: No

Reviewer #4: No

---

## [Decision Letter · Decision Letter 2]

12 Nov 2025

Investigating the dose-response relationship between music and anxiety reduction: A randomized clinical trial

PMEN-D-24-00371R2

Dear Ms. Mullen,

We are pleased to inform you that your manuscript 'Investigating the dose-response relationship between music and anxiety reduction: A randomized clinical trial' has been provisionally accepted for publication in PLOS Mental Health.

Best regards,

Haroon Lone

Academic Editor

PLOS Mental Health

Reviewer Comments (if any, and for reference):

Reviewer's Responses to Questions

**Comments to the Author**

Reviewer #3: All comments have been addressed

Reviewer #4: (No Response)

publication criteria?

Reviewer #3: Yes

Reviewer #4: (No Response)

3. Has the statistical analysis been performed appropriately and rigorously?

Reviewer #3: Yes

Reviewer #4: (No Response)

4. Have the authors made all data underlying the findings in their manuscript fully available (please refer to the Data Availability Statement at the start of the manuscript PDF file)?

Reviewer #3: Yes

Reviewer #4: (No Response)

5. Is the manuscript presented in an intelligible fashion and written in standard English?

Reviewer #3: Yes

Reviewer #4: (No Response)

Reviewer #3: All of my comments are addressed accordingly by the author. The manuscript contains interesting topic but also heavy with information, terms and details which need to be presented and articulated properly as well as consistently.

Reviewer #4: All my concerns are addressed.

what does this mean?). If published, this will include your full peer review and any attached files.

**Do you want your identity to be public for this peer review?** For information about this choice, including consent withdrawal, please see our Privacy Policy

Reviewer #3: No

Reviewer #4: No
